# Augmenting a prognostic deep learning system for referable diabetic retinopathy and maculopathy with synthetic retinal images

Paul Nderitu [1,2] ✉, Joan M. Nunez do Rio [1], Laura Webster[3], Samantha S. Mann [3,4], David Hopkins[5,6], Christos Bergeles [7] & Timothy L. Jackson[1,2]

## Abstract

**Background** Labelled data scarcity and class imbalance are common deep learning system (DLS) development challenges. We investigated if synthetic retinal images from a conditional cascaded diffusion model (CCDM) improves prognostic DLS (pDLS) performance for 2-year incident referable diabetic retinopathy or maculopathy (rDR/rM) prediction.

**Methods** Macula images from 72,559 eyes (September 2013 to December 2019) from the UK South-East London Diabetic Eye Screening Programme (SEL-DESP) formed the development dataset, whilst 9,071 eyes were used for internal testing. Images from 2,842 eyes from Birmingham DESP were used for external testing. Prognostic DLS were augmented with ×1, ×2, and ×4 additional synthetic positive cases (pDLS-G) and compared to unaugmented (pDLS-N) and ×1 positive-case resampled pDLS (pDLS-R) using the Area-Under-the Receiver Operating Characteristic curve (AUROC).

**Results** Here we show that CCDM generate realistic synthetic retinal images that are comparable to real images and demonstrate the utility of synthetic retinal images in augmenting the development of a pDLS. The internal and external test AUROC for the pDLS are 0.827 (95% CI: 0.794–0.861) and 0.756 (0.680–0.831), respectively. Augmentation with ×2 additional synthetic positive cases (pDLS-G ×2) significantly improves the internal test AUROC to 0.845 (95% CI: 0.812–0.877, $p = 0.044$) but does not improve the external test AUROC 0.717 (0.633–0.828, $p = 0.243$). Resampling positive real cases alone does not improve pDLS-R performance.

**Conclusions** Augmenting pDLS with synthetic retinal images significantly improves pDLS performance on internal testing but not external testing suggesting further research is required to enhance the generalisability of synthetic retinal image augmentation.

## Plain Language Summary

Artificial Intelligence (AI) can be applied to images of the eyes to predict the presence of diabetic eye complications. However, setting up AI systems that enable this requires labelled images. This study explored whether adding images from the back of the eye (retina) could improve predictive AI model accuracy in identifying diabetic eye disease. We compared whether AI models trained using real images and images generated by computers could be used to predict future eye complications. Images generated by computers had similar features to real images and improved predictive AI model accuracy. The key finding is that synthetic retinal images can enhance predictive AI model performance. This could mean that AI created with the help of synthetic retinal images would be better at predicting diabetic eye disease, and this could be used as an accurate tool to assist clinicians to better monitor and treat the disease.

A common challenge when developing deep learning systems (DLS) for healthcare is the scarcity of high-quality labelled data[1]. An additional challenge is that many diseases of interest have a low incidence or prevalence with limited positive cases, resulting in imbalanced development datasets.

Training DLS with imbalanced datasets can result in limited model performance and generalisation due to the paucity of positive cases[1]. Approaches to managing class imbalance include over or under sampling available data, or data augmentation[1]. Recent studies suggest that synthetic retinal

[1]Section of Ophthalmology, Faculty of Life Science and Medicine, King's College London, London, UK. [2]King's Ophthalmology Research Unit, King's College Hospital, London, UK. [3]South East London Diabetic Eye Screening Programme, Guy's and St Thomas' Foundation Trust, London, UK. [4]Department of Ophthalmology, Guy's and St Thomas' Foundation Trust, London, UK. [5]Department of Diabetes, School of Life Course Sciences, King's College London, London, UK. [6]Institute of Diabetes, Endocrinology and Obesity, King's Health Partners, London, UK. [7]School of Biomedical Engineering & Imaging Sciences, King's College London, London, UK. ✉e-mail: p.nderitu@doctors.org.uk

images can be used to augment, or even replace real retinal images when training discriminative DLS[2]. Therefore, synthetic images offer a potential solution to addressing label scarcity and dataset imbalance[3]. Synthetic data also have the additional benefit that they could be used to overcome data governance and sharing concerns, since synthetic data is considered a form of anonymised data[3].

Generative models can be trained to synthesise new data instances that closely resemble real training data examples. Generative approaches for images include variational autoencoders[4], autoregressive networks[5], generative adversarial networks (GANs)[6], and recently, diffusion models[7,8]. Diffusion models have become the de facto method for photorealistic image generation, outperforming previously incumbent GANs[9]. Two common diffusion-based approaches for high-quality image generation are using a conditional cascaded diffusion model (CCDM), or a latent diffusion model (LDM)[8,10]. A CCDM is a stacked diffusion model which sequentially synthesises larger images conditioned on text and a prior lower-scale generated image[10]. GANs have been used to generate realistic retinal images to train discriminant DLS to detect diabetic retinopathy (DR)[11–14]. However, few studies have explored the use of diffusion models to generate synthetic images of DR[15], and none have assessed their utility in a prognostic use case such as predicting DR progression.

The UK national diabetic screening programme (DESP) is attended by 3.2 million individuals annually. The purpose of DESP is to detect incident referable DR (rDR) or referable maculopathy (rM) using 2-field retinal images, the presence of which requires closer monitoring or treatment. Therefore, predicting incident rDR/rM is of clinical importance as it could allow for earlier referable DR detection, enabling individualised screening and potentially preventive interventions[16]. However, the 2-year incidence of rDR/rM is low at round 2% making the development of prognostic DLS using screening data challenging and an imbalanced task[16]. The use of additional synthetic cases of incident referable disease would reduce the degree of class imbalance when training a prognostic DLS which could improve the performance of the predictive model[16–20].

In this study, a CCDM is trained to generate synthetic retinal images conditioned on clinicodemographic characteristics of age group, sex, self-declared ethnicity, diabetes duration group, presence/absence of baseline mild DR, and presence/absence of 2-year incident rDR/rM. A trained CCDM is used to synthesise retinal images of positive cases (2-year incident rDR/rM) to augment a development dataset of real retinal images. We assess the effect of augmenting the development dataset with additional synthetic positive cases to reduce class imbalance for a prognostic DLS (pDLS) that predicts 2-year incident rDR/rM. Synthetic image quality is quantitatively compared to real images using the Fréchet Inception Distance (FID). The effect of CCDM guidance using the clinicodemographic characteristics is also qualitatively assessed using human grader and retinal expert evaluations. Finally, Uniform Manifold Approximation and Projection (UMAP) are applied to embeddings from real and synthetic retinal images to qualitatively assess for similarity.

We show that synthetic positive cases generated by a CCDM significantly improve the internal prognostic performance of a pDLS in predicting 2-year incident rDR/rM, demonstrating the CCDM's ability to create realistic, clinicodemographically-conditioned retinal images. However, performance benefits from synthetic retinal image augmentation does not generalise to external testing, indicating a need for improve strategies for synthetic retinal image augmentation.

## Methods
### Study population and datasets
Data from 203,983 eyes of 102,601 individuals from the South-East London DESP (SEL-DESP) and 9444 eyes of 4778 individuals from Birmingham DESP (B-DESP) between Sept 2013 to Dec 2019 were extracted. Included eyes were those with a baseline visit and final visit spaced 2 years apart ±4 months. The baseline retinal image (macula field) and characteristics of age, sex, self-declared ethnicity (White, Black, South Asian, Other Asian, Mixed, Not specified), diabetes duration, and baseline/final visit DR and

maculopathy grades (see Supplementary Table 1 for grading definitions and Supplementary Table 2 for a summary of screening procedures) were extracted for each included eye. Age was categorised into 12–30, 31–50, 51–70, 71–90 and ≥91 year groups, whilst diabetes duration was categorised in 0–10, 11–20, 21–30, ≥31 year groups. Age and diabetes duration were categorised post-hoc from continuous values to provide a more consistent, discrete conditioning signal for the CCDM, but using clinically relevant intervals. Missing variables were categorised as 'Not Specified', and these eyes were also included. After selection, data from 72,559 eyes from SEL-DESP (80%) were used for CCDM and pDLS development, 9071 SEL-DESP eyes (one eye selected at random per patient) were used for CCDM and pDLS internal testing, and 2842 B-DESP eyes (one eye selected at random per patient) were used for pDLS external testing (Fig. 1). The right and left eyes of the same patient were used either in the training or testing sets but not both.

### Statistics and reproducibility
**CCDM development and synthetic retinal image generation**. A CCDM akin to IMAGEN[10] with a design optimised for training and sampling was used[21]. The CCDM was trained to generate both positive (2-year incident rDR/rM present) and negative cases (2-year incident rDR/rM absent) (Fig. 1). The CCDM consisted of a base model which learned to generate a 64×64 resolution pixel macula image conditioned on the trainable embeddings of clinicodemographic characteristic of age group, sex, self-declared ethnicity, diabetes duration group, presence/absence of baseline mild DR and presence/absence of 2-year incident rDR/rM. A subsequent super-resolution model conditioned on the lower resolution 64 × 64 pixel image and embeddings of the clinicodemographic characteristic was concurrently trained to generate the final 512 × 512 pixel macula image (Fig. 2). CCDM were trained on two P6000 24GB NVIDIA GPUs with a total training time of 9 days (~50 epochs). The time taken to generate 9071 test set images with a batch size of 8 was 9 h and 50 min on a single P6000 (~31 s/iteration). The use of conditioning to allow for the fine-grained control of image synthesis increased training time by ~20% and doubled the inference time[7]. Further details on CCDM development and hyperparameters are provided in the *supplementary methods*.

**Prognostic DLS model development**. An ImageNet initialised EfficientNet-V2-s model was used as the base pDLS which was trained to predict 2-year incident rDR/rM with auxiliary tasks of predicting 2-year incident rDR alone, 2-year incident rM alone and the presence of baseline mild DR (Fig. 1). The pDLS training datasets consisted of (1) an unaugmented development dataset (pDLS-N), (2) an augmented development dataset with double the number of positive cases of the unaugmented dataset achieved via ×1 resampling of real positive cases (pDLS-R), (3) an augmented development dataset with ×1, ×2 and ×4 additional synthetic positive cases generated by the trained CCDM (pDLS-G). We aimed to reduce the degree of class imbalance by a factor of ~5 which would incur a modest CCDM sampling computational cost compared to generating enough positive cases to match every negative case to get a full a fully balanced dataset. The pDLS were optimised using the loss function described in Eq. 1. Further details on pDLS development and hyperparameters are provided in the *supplementary methods*.

$$\text{Loss}_{\text{long}} = \text{Loss}_{\text{progression}} + \text{Loss}_{\text{auxillary}} + \lambda ||\theta||_2 \quad (1)$$

$$\text{Loss}_{\text{progression}} = -\frac{1}{N} \sum \sum_c^C [w_{P_c+} P_c \log(\hat{P}_c) + (1 - P_c) \log(1 - \hat{P}_c)]$$

$$\text{Loss}_{\text{auxillary}} = -\frac{1}{N} \sum [w_{B+} B \log(\hat{B}) + (1 - B) \log(1 - \hat{B})]$$

N = No. of samples, C = No. classes, $w_{P_c+}$ = Positive incident disease case weighting, P = Incident disease ground truth, $\hat{P}$ = Incident disease

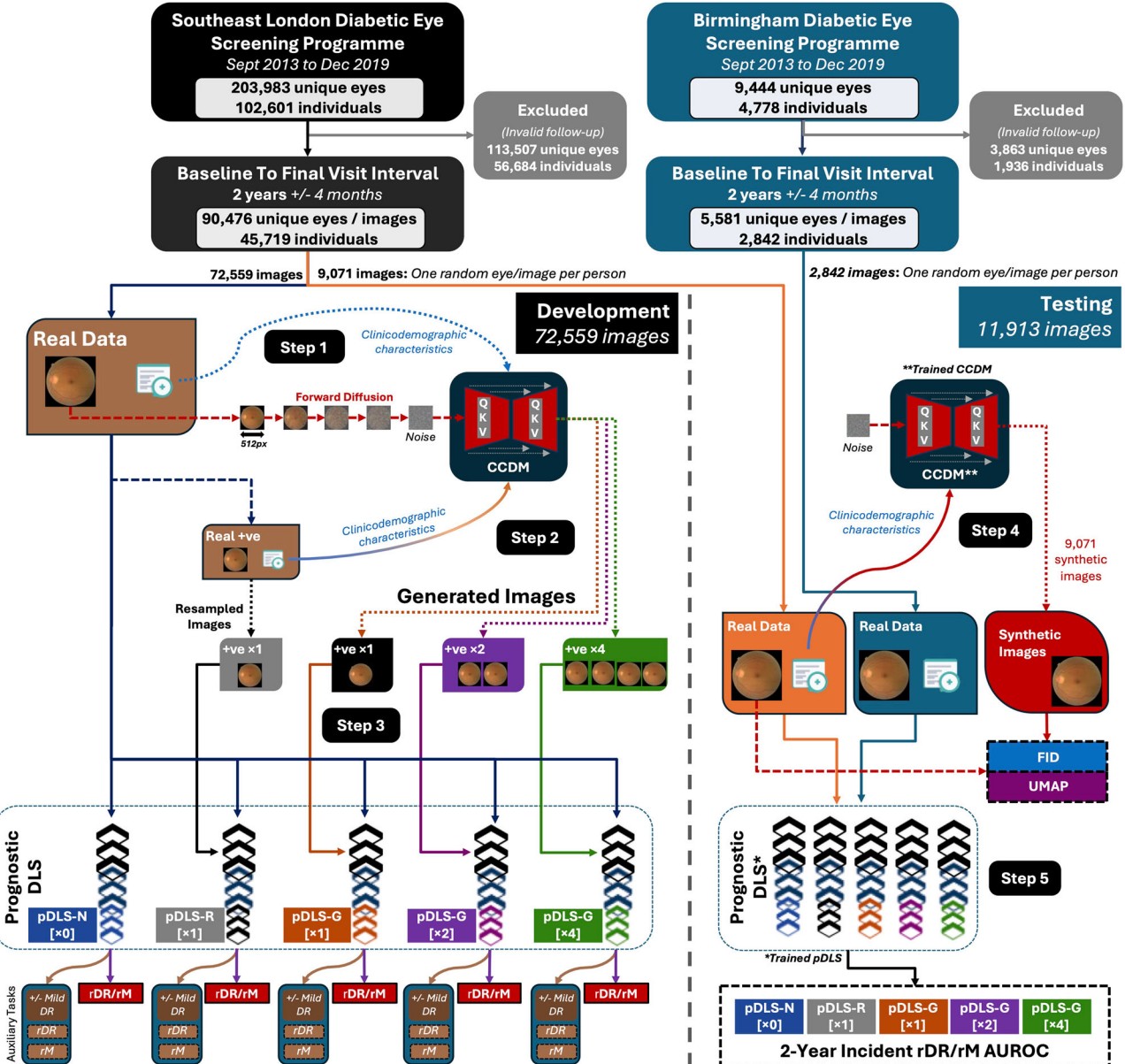

**Fig. 1 | Study dataset flowchart, model development, testing and outcomes. Step 1:** The real macula images consisting of positive and negative samples (72,559 images) are used to train the CCDM conditioned on clinicodemographic characteristics. CCDM learn to reverse the forward diffusion process to generate new samples. **Step 2:** The trained CCDM is used to generate images of positive cases conditioned on the clinicodemographic variables of real positive samples to generate x1, x2 and x4 additional positive cases. **Step 3:** Four prognostic DLS (pDLS) are then trained to predict incident referable DR and maculopathy using all the real positive and negative samples for the native pDLS (pDLS-N), x1 additional resampled positive cases (pDLS-R), x1 additional generated positive cases (pDLS-G x1), x2 additional generated positive cases (pDLS-G x2), and x4 additional generated positive cases (pDLS-G x4). **Step 4:** The trained CCDM is used to generate

9071 synthetic images using the clinicodemographic characteristics of the real internal test set. The image embeddings of the real internal test set retinal images are compared to the corresponding synthetic images using the FID and UMAP. **Step 5:** Images from the internal and external test sets (9071 and 2842 images respectively) are then used to evaluate the performance of the pDLS to assess its performance in predicting incident referable DR and maculopathy. CCDM Conditional cascaded diffusion model, pDLS Prognostic Deep Learning System, +ve Positive cases, rDR Referable diabetic retinopathy (2-year incident), rM Referable maculopathy (2-year incident), QKV Attention mechanism, AUROC Area-under-the receiver operating characteristic, N Native, R Resampled, G Generated, px Pixels, FID Fréchet Inception Distance, UMAP Uniform Manifold Approximation and Projection (UMAP).

prediction, $w_{B+}$ = Positive auxiliary case (baseline mild DR present) weighting, B = Auxiliary task ground truth, $\hat{B}$ = Auxiliary task prediction, $\lambda$ = Regularisation parameter, $\theta$ = Model parameters.

**CCDM outcomes.** We conducted a human grader test where 50 real and 50 synthetic macula images were randomised and assessed by a senior and junior human grader from SEL-DESP. Their tasks were to grade: (1) The level of DR, and (2) classify images as real or generated. Graders were

blinded to the proportion of real versus generated images and the level of DR (R0, R1, R2 or R3A) or maculopathy (M0 or M1). However, all images in the test had either no DR (R0) or mild DR (R1), and none had maculopathy (M0). Graders were also asked to provide the subjective criteria they devised to differentiate between real and generated images at the end of the test. An additional qualitative assessment of 18 uncurated synthetic images of positive and negative cases from the CCDM was also performed by a retinal specialist (PN).

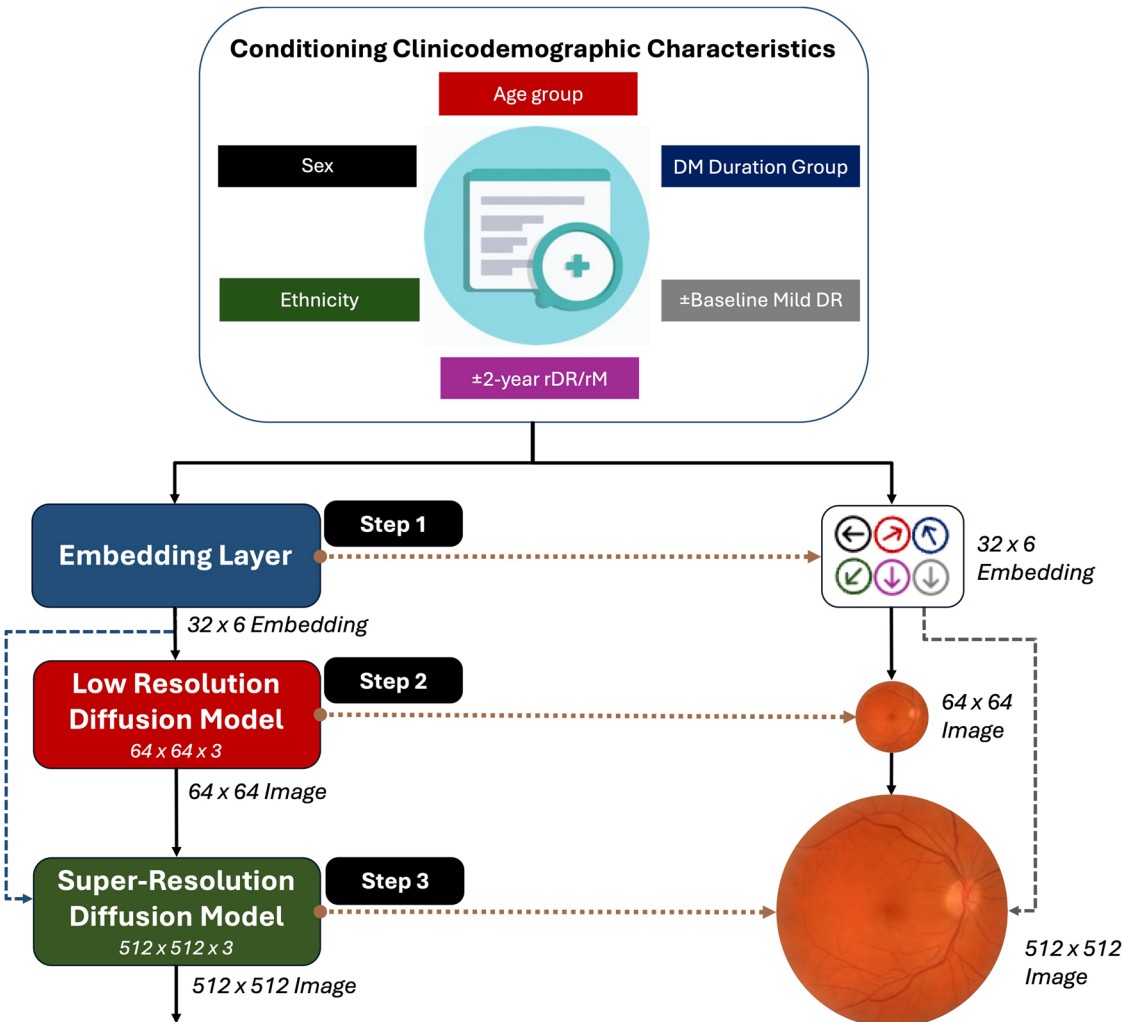

**Fig. 2 | CCDM architecture. Step 1:** The clinicodemographic categorical characteristics are converted into a numerical vector embedding of length 32 representing each of the 6 characteristics (embedding shape 32 × 6). **Step 2:** These embedding are used to condition a CCDM which learn to generate a low-resolution synthetic retinal image of size 64 × 64 pixels. **Step 3:** The low-resolution synthetic retinal image and clinicodemographic characteristics are used to condition the second super resolution stage of the CCDM to generate the final 512 × 512 resolution synthetic retinal image. DM Diabetes Mellitus, rDR Referable diabetic retinopathy, rM Referable maculopathy.

CCDM synthetic image quality was quantitatively evaluated using the FID which estimates the similarity between real and synthetic retinal image feature distributions[22]. Both positive and negative synthetic cases were generated using clinicodemographic characteristics of real internal test set (Fig. 1). Additionally, unsupervised UMAP (nearest neighbours 15) was used to qualitatively assess the similarity between embeddings from 9071 real and 9071 synthetic retinal images from the internal test set. Embeddings were extracted using the RETFound foundation model for colour fundus photographs[23].

**pDLS outcomes.** Internal (SEL-DESP) and external (B-DESP) test datasets, which only included real retinal images, were used to compute the 2-year incident rDR/rM AUROC and AUPRC to assess the performance of the pDLS.

**Statistical analysis.** Analyses were performed between October 2024 and March 2025 using Python v3.9. Confidence intervals and statistical differences for the AUROC were determined using the Delong method and Delong's test respectively comparing the augmented DLS (pDLS-G or pDLS-R) with the unaugmented DLS (pDLS-N) as the ref. 24 A $p < 0.05$ was considered statistically significant.

**Ethics and approvals**

The study was conducted in accordance with the tenets of the declaration of Helsinki and TRIPOD guidelines. UK health research authority approval and a favourable ethical opinion from the UK east midlands Leicester south research Ethics committee (REC) were attained prior to study commencement (20/EM/0250, 6/October/2020). Licensed access for the external dataset (INSIGHT UK) was approved by the west of Scotland REC 4 (20/ES/0087). The need for informed consent was waived by the favourable ethical opinion. Study data were anonymised prior to extraction, and individuals who previously objected to the use of their data for research were excluded.

**Results**

**Baseline characteristics**

The baseline characteristics of included individuals and eyes, including positive cases in the development set are shown in Table 1. Development and internal test set baseline characteristics for were similar, but the external test population differed in ethnicity group (Black 27.7% vs 5.7% and South Asian 6.5% vs 22.7%) and mean diabetes duration (8 years vs 6 years).

## Table 1 | Baseline characteristics

| Characteristics | | SEL-DESP | | | B-DESP |
|---|---|---|---|---|---|
| | | Development Set 72,559 unique eyes 36,648 individuals | | Internal Test Set 9071 unique eyes 9071 individuals | External Test Set 2842 unique eyes 2842 individuals |
| | | N (%) or [mean] (SD) | | | |
| | | All Cases | *Positive Cases Only* | All Cases | All Cases |
| Age (years) | | [61] (14) | [56] (15) | [62] (14) | [64] (14) |
| Sex | Female | 16,837 (45.9) | 489 (43.3) | 4126 (45.5) | 1330 (47.6) |
| | Male | 19,810 (54.0) | 640 (56.7) | 4945 (54.5) | 1442 (51.0) |
| | Not Specified | 1 (0.1) | 0 (0.0) | 0 (0.0) | 70 (1.4) |
| Ethnicity | White | 19,423 (53.0) | 487 (43.1) | 4792 (52.8) | 1698 (59.7) |
| | Black | 10,137 (27.7) | 435 (38.5) | 2487 (27.4) | 163 (5.7) |
| | South Asian[a] | 2396 (6.5) | 73 (6.5) | 631 (7.0) | 646 (22.7) |
| | Other Asian[b] | 2380 (6.5) | 60 (5.3) | 600 (6.6) | 57 (2.0) |
| | Mixed | 893 (2.4) | 31 (3.8) | 250 (2.8) | 28 (1.0) |
| | Other | 929 (2.5) | 23 (2.0) | 198 (2.2) | 17 (0.6) |
| | Not Specified | 490 (1.3) | 20 (1.8) | 113 (1.3) | 233 (8.2) |
| Diabetes Duration (years) | | [8.0] (7.0) | [11.00] (7.6) | [8.1] (6.9) | [6.0] (3.5) |
| Baseline Mild DR Present (*eyes*) | | 10,373 (14.3) | 880 (65.1) | 2519 (14.1) | 366 (12.9) |
| 2-Year Incident Referable DR (*eyes*) | | 232 (0.3) | *232 (17.17)* | 58 (0.3) | 11 (0.4) |
| 2-Year Incident Referable Maculopathy (*eyes*) | | 1235 (1.7) | 1235 (91.41) | 275 (1.5) | 48 (1.7) |

*SEL-DESP* South-East London diabetic eye screening programme, *B-DESP* Birmingham diabetic eye screening programme, *SD* Standard deviation, *DR* Diabetic retinopathy.
[a]Incudes Indian, Pakistani and Bangladeshi ethnicity groups.
[b]Includes Chinese ethnicity group.

### CCDM synthetic retinal image quality quantitative and qualitative analysis

After training, 9071 synthetic retinal images were generated using the CCDM conditioned on internal test set clinicodemographic characteristics (Fig. 1).

**Human grader test**. The senior grader achieved 100% accuracy in differentiating between real and generated retinal images. In contrast, the junior grader attained an overall accuracy rate of 62%, which included a 96% accuracy for real images and a 28% accuracy for generated images. The graders' accuracy in distinguishing between no DR and mild DR was 87.5%, comprising 95% accuracy for real images and 82% for generated images. Both graders accurately recognised the absence of maculopathy in all real and generated cases. Graders felt that synthetic images had unnaturally straight or angular blood vessels, fine vessels in periphery, and illogical vessel branching.

**Quantitative and qualitative assessments**. The FID achieved was low at 9.3 (lower is better) demonstrating a high similarity between the distribution of synthetic and real retinal image features. Synthetic retinal images demonstrating the effects of changing the clinicodemographic characteristics of age, sex, ethnicity, and diabetes duration are shown in Fig. 3. An additional 18 uncurated examples of positive and negative cases with randomly varied clinicodemographic characteristics are shown in Supplementary Fig. 1, demonstrating the diversity of the generated synthetic retinal images. A qualitative assessment of synthetic retinal images showed that they faithfully reproduced the major features of real retinal images including the optic disc, optic cup, fovea, retinal vasculature, choroidal markings, black circular image mask, and laterality mask notch. Synthetic retinal images showed an increasing degree of reflectivity, orange colouration and clarity when conditioned using a younger age group which is clinically consistent with the appearance of the retinal appearance of younger individuals. Conversely, increasing age group conditioning resulted in synthetic retinal images which were more hazy mimicking the effects of a developing cataract[25]. Structural changes were less evident subjectively when varying sex. Increasing diabetes duration in isolation had limited effects on the synthetic retinal images. Black or South Asian ethnicity group conditioning resulted in synthetic images with greater retinal pigmentation in keeping with correlations between retinal pigmentation and ethnicity[26]. Some generated macula images demonstrated irregular vessel branching. As shown in Fig. 4, the unsupervised UMAP of real and synthetic retinal image embeddings showed a high degree of similarity suggesting the synthetic retinal images likely had similar features to real retinal images.

### Prognostic DLS performance

Native pDLS (pDLS-N) 2-year incident rDR/rM AUROC were 0.827 (95% CI: 0.794–0.861) on internal testing and 0.756 (0.680–0.831) on external testing (Table 2). Augmenting the pDLS development dataset with additional synthetic positive cases (pDLS-G) increased internal test 2-year incident rDR/rM AUROC compared to the native pDLS at all ratios (×1, ×2, ×4). There was a statistically significant improvement in the AUROC to 0.845 (0.812–0.877, $p = 0.044$) with ×2 additional synthetic positive cases on internal testing (Table 2). Additionally, there was an improvement in the Area-Under-the Precision Recall Curve (AUPRC) by +0.032 on internal testing and +0.056 on external testing (Supplementary Table 3). However, pDLS-G AUROC did not improve on external testing, although there were consistent AUPRC improvements. Dataset augmentation using real positive case resampling (pDLS-R) resulted in a non-significant reduction in both the AUROC and AUPRC.

### Discussion

In this study, we developed a CCDM capable of generating realistic synthetic retinal images conditioned on clinicodemographic variables. The CCDM achieved a low FID of 9.3 on the internal test set with similar UMAP embeddings, suggesting synthetic retinal images closely matched the feature distribution of real retinal images. Although evaluation datasets differ, the FID was better than previously reported for synthetic retinal images

**Fig. 3 | Effect of clinicodemographic conditioning on synthetic retinal image generation.** This figure shows the effect of the clinicodemographic characteristic conditioning on CCDM synthetic retinal image generation. Only one clinicodemographic variable is changed at a time with other conditioning variables kept constant except for diabetes duration group whereby the commensurate age group (starting at 51–70 y) is increased with increasing diabetes duration group. The image shows synthetic retinal image changes which are clinically plausible for age (e.g. due to cataract development). Sex does not subjectively change the synthetic retinal image morphology. Ethnicity changes are reflected in the degree of retinal pigmentation but also some laterality changes are noted. Diabetes duration has a modest effect on synthetic retinal image clarity. The noise image is kept constant at each generation, and stochastic CCDM sampling parameters (e.g. sigma churn) are set to deterministic values to ensure only the clinicodemographic characteristics affect the synthetic retinal image generation. rDR=Referable diabetic retinopathy, rM=Referable maculopathy. *The default clinicodemographic characteristics are age group 51–70, sex female, ethnicity white, diabetes duration group 0–10 years, baseline mild DR absent and 2-year incident rDR/rM absent.

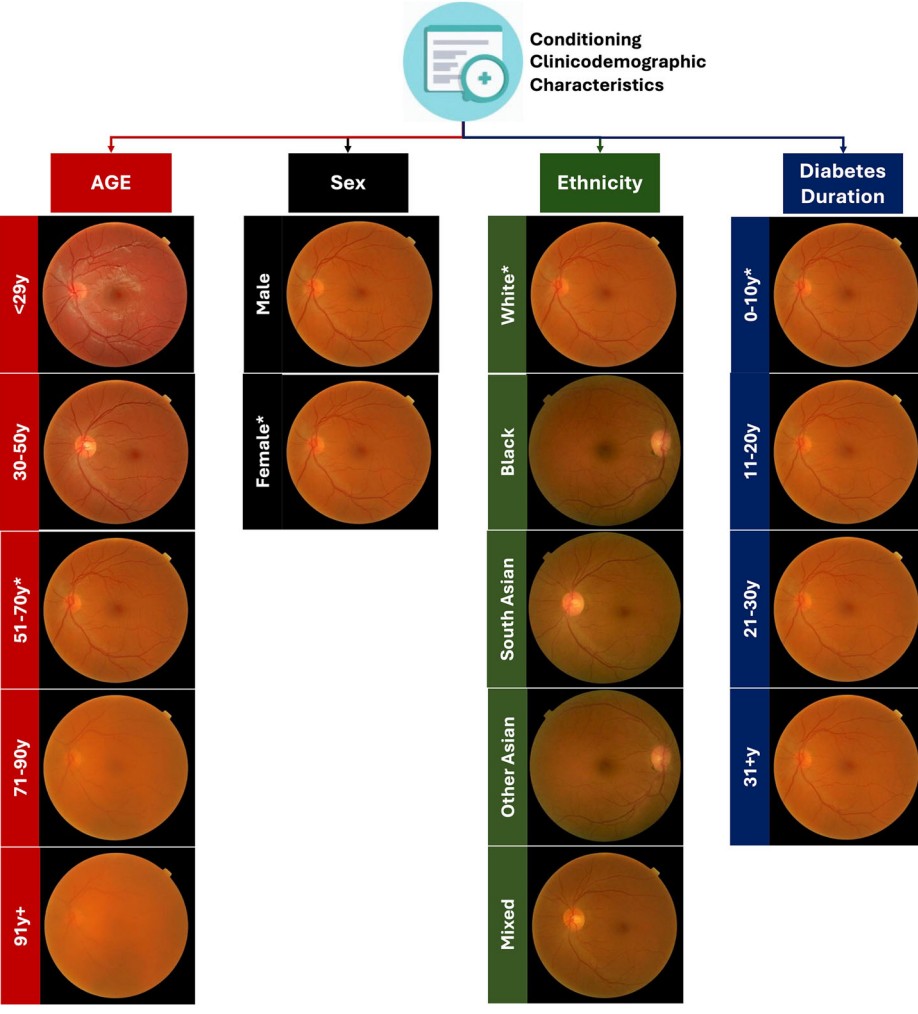

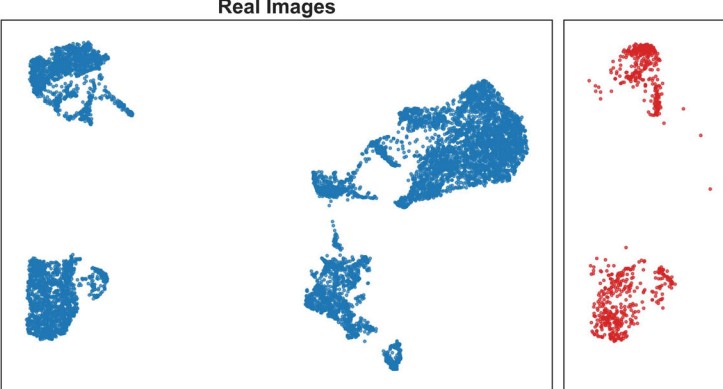
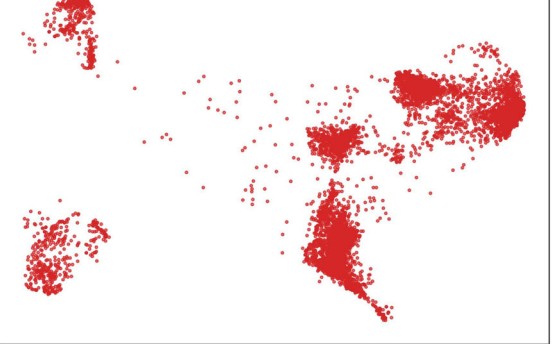
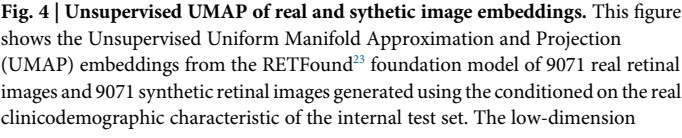

**Fig. 4 | Unsupervised UMAP of real and sythetic image embeddings.** This figure shows the Unsupervised Uniform Manifold Approximation and Projection (UMAP) embeddings from the RETFound[23] foundation model of 9071 real retinal images and 9071 synthetic retinal images generated using the conditioned on the real clinicodemographic characteristic of the internal test set. The low-dimension embeddings of real and synthetic images show a high degree of similarity suggesting that real and synthetic retinal image features are also similar. Images were resized to 224 × 224 prior to embedding as the required image size for RETFound and the Euclidean metric and space was used for the UMAP embeddings.

generated by diffusion models (48.5 to 86.8)[27–29], and similar to the best reported FID for GANs (4.2)[11].

Although quantitative and qualitative metrics were favourable, evaluation by human graders revealed that the senior graders was generally able to distinguish synthetic images from real images, whereas a junior grader more frequently misclassified generated images as real. This contrasts with findings by a previous study by Kim et al., which reported no difference in real vs generated grading accuracy based on human grader experience[30]. A key distinction between their study and our research is that they used GAN-generated images and a training set composed primarily of normal eyes[30].

**Table 2 | Prognostic DLS performance**

| Prognostic DLS [Additional Positive Cases] | Positive & Negative Training Cases Ratio [Samples][a] | 2 Year Incident Disease Outcomes | SEL-DESP (Real Internal Test Set) N 9071 unique eyes / retinal images | | | B-DESP (Real External Test Set) N 2842 unique eyes / retinal images | | |
|---|---|---|---|---|---|---|---|---|
| | | | AUROC (95%CI) | Δ vs pDLS-N | p vs pDLS-N[b] | AUROC (95%CI) | Δ vs pDLS-N | p vs pDLS-N[b] |
| pDLS-N [×0] | 1:58 [1351: 71,208] | rDR/rM | 0.827 (0.794–0.861) | - | - | 0.756 (0.680–0.831) | - | - |
| pDLS-R [×1] | 1:26 [2702: 71,208] | rDR/rM | 0.823 (0.788–0.857) | −0.004 | 0.671 | 0.718 (0.643–0.794) | −0.038 | 0.109 |
| pDLS-G [×1] | 1:26 [2702: 71,208] | rDR/rM | 0.847 (0.816–0.877) | +0.020 | 0.079 | 0.761 (0.689–0.833) | +0.005 | 0.827 |
| pDLS-G [×2] | 1:19 [4053: 71,208] | **rDR/rM** | **0.851 (0.820–0.882)** | +0.024 | **0.044*** | 0.717 (0.633–0.828) | −0.039 | 0.243 |
| pDLS-G [×4] | 1:11 [6755: 71,208] | rDR/rM | 0.844 (0.812–0.875) | +0.017 | 0.145 | 0.750 (0.673–0.828) | −0.006 | 0.840 |

**Bold** = AUROC significant improvement compared to pDLS-N. Underlined=Numerical improvement compared to pDLS-N.
AUROC Area-under-the Receiver Operating Characteristic curve, 95%CI 95% Confidence Interval, SEL-DESP Southeast London diabetic eye screening programme, B-DESP Birmingham diabetic eye screening programme, Δ Difference, DR Diabetic Retinopathy, rDR/rM Referable diabetic retinopathy or maculopathy (2-year incident), pDLS-N Native unaugmented image prognostic DLS, pDLS-R Resampled positive case augmented prognostic DLS, pDLS-G Generated positive case augmented prognostic DLS.
[a]Ratios and number of positive and negative cases (eyes) used during pDLS training.
[b]Significant if p < 0.05.

However, in concordance with our study, Kim et al. also reported that graders reported inconsistencies in retinal vasculature in generated retinal images[30]. Interestingly, one grader remarked that certain generated images elicited a queasy, uneasy feeling, akin to the 'uncanny valley' effect noted with virtual faces[31]. Real images were more representative of the true DR level compared to generated images which were consistently graded as showing no retinopathy when conditioned to generate mild DR. This may be due to challenges in the CCDM synthesising small DR lesions like microaneurysms.

The pDLS augmented by ×2 additional positive cases (pDLS-G x2) significantly outperformed the unaugmented pDLS-N on internal testing but not external testing. These findings suggest augmenting the development dataset using synthetic retinal images may improve prognostic DLS performance on imbalanced datasets, including for predictive tasks, where cases are similar to the training population. The optimal ratio of synthetic to real retinal images likely varies and requires exploration during DLS development on a task-by-task basis. Further refinement of CCDM training or sampling strategies is also required to ensure downstream DLS generalise to external populations when augmenting development datasets with synthetic retinal images.

Few studies have developed diffusion models that can generate synthetic colour retinal images[27–29,32], with the majority of prior studies using GANs[12–14,30,33–39]. Uniquely, we developed a CCDM which generates synthetic retinal images conditioned on clinicodemographic characteristics without requiring conditioning on the retinal vascular tree or DR lesion segmentation masks[11–14,28,29,33,35–37,39]. The CCDM generated synthetic retinal images that realistically reproduced important clinicodemographic image features. This suggest that CCDM and diffusion models could be used to generate synthetic retinal image datasets representative of populations with specific characteristics (e.g. ethnicity distributions) to enrich datasets where such populations are underrepresented. This is an important finding because synthetic retinal images could rebalance DLS development datasets where ethnic groups are underrepresented, in turn improving the model's performance amongst these individuals. It may be possible to also use other clinical variables, such as biochemical values (e.g. HbA1c) or medications, as conditioning to assess how synthetic retinal image features are affected. This could provide clues about possible interactions between the conditioning variables and retinal image features. Generative models could therefore be used as an exploratory tool to generate hypotheses on plausible interactions between conditioning variables and retinal morphology[40]. For example, if a medication with unknown clinical effects on the retina is used as conditioning, generated images could be used to examine the plausible effects of the medication on retinal morphology. It may also be possible to explore the diversity of the relationships between conditioning variables and retinal morphology, for example the variation that exists between ethnicity and retinal pigmentation[26]. Unlike previous studies which used cross-sectional variables, our approach also used a prognostic risk factor (2-year incident rDR/rM) as conditioning to guide synthetic retinal image generation. To best of our knowledge, the use of a longitudinal variable as conditioning for retinal image generation has not been previously reported. Using longitudinal conditioning variables could allow for the generation of plausible retinal images with features representing events at arbitrary future or historical time points. Generating longitudinal counterfactuals could be useful for prognostication or to influence patient behaviour, such as encourage better diabetes control by showing individuals synthetic retinal images emulating the development sight-threatening DR.

Few studies have assessed the downstream effects on DLS performance of augmenting development datasets with synthetic retina images[2,11–13,15]. A study by Veturi et al. (2022) used GANs to generate synthetic fundus autofluorescence images representing different inherited retinal diseases[2]. They reported synthetic autofluorescence images could wholly substitute real images when training a discriminant DLS, achieving equivalent performance to a DLS trained on real autofluorescence images[2]. However, augmenting real autofluorescence images with additional synthetic autofluorescence images did not significantly improve discriminant DLS

performance[2]. A recent study also reported improved vision foundation model performance when combining a specific ratio (1:5) of real and synthetic slit-lamp ophthalmic images[41]. Five studies have reported improved DR detection after augmenting discriminant DLS datasets with synthetic retinal images on internal testing, with all but one study using GANs[11-15].

In our study, pDLS augmented with ×2 additional synthetic positive cases (pDLS-G) showed significantly better prognostic performance in predicting 2-year incident rDR/rM on internal testing. Although not reaching statistical significance, all pDLS-G outperformed prognostic models augmented with ×1 additional resampled real positive cases (pDLS-R) on internal testing. This suggests that synthetic retinal images of positive cases improve prognostic model performance compared to the classic resampling augmentation strategy. However, the pDLS augmented with additional synthetic positive cases did not show better AUROC performance on external testing, although there were consistent improvements to the AUPRC. The difference in pDLS performance between internal and external testing suggests that synthetic retinal images, which share the distribution of the development dataset, conferred an advantage to pDLS performance when populations were similar, but this did not consistently generalise to external populations. The distribution of features and populations of positive cases may differ between internal and external populations as there were differences in baseline characteristics between these populations (see Table 1).

This study has some limitations. The sampling time required to generate synthetic retinal images restricted experiments to generating positive cases instead of the larger volume of negative cases. However, positive samples were those which were significantly underrepresented and were the focus of the imbalanced use case. Computational constraints restricted experiments to single-field synthetic macula images as opposed to 2-field retinal images captured in UK DR screening. However, this constraint enabled the isolated evaluation of synthetic macula image augmentation. Conditioning was achieved using a simple embedding of categorical clinicodemographic variables, but more complex caption-based conditioning using pre-trained CLIP embeddings or large language encoder models could provide more nuanced control during synthetic retinal image generation of more subtle features such as exudates and microaneurysms[42]. Additional experiments to assess the effectiveness of clinicodemographic conditioning using trained DLS classifiers could also be explored although the majority of baseline images in our predictive use case had no retinopathy (~85%).

## Conclusions

In conclusion, CCDM generated synthetic retinal images with realistic clinicodemographic characteristics. CCDM can be used to synthesise retinal images to augment the development dataset of prognostic DLS to reduce class imbalance and potentially improve predictive DLS performance. Augmenting development datasets with synthetic retinal images showed encouraging improvements in prognostic DLS performance in predicting 2-year incident rDR/rM on internal testing, but improvements did not generalisation on external testing. Future studies should assess the effect of using larger parameter diffusion models, more complex generative model conditioning, and improving the generalisation of downstream DLS trained on development datasets augmented with synthetic retinal images.

## Data availability

De-identified data (retinal images and clinicodemographic risk factor variables) used for development and internal testing are not publicly available at present due to the absence of authorisation for public data sharing. However, the data used for external validation can be requested from INSIGHT with access/approval dependant on meeting data governance and application criteria. The source data for Fig. 4 is in Supplementary Data 1.

## Code availability

Open-source resources and code were used to develop and validate the prognostic deep learning systems including python, pytorch, and timm. The

code for the generative CCDM is also open access and available at https://github.com/pnderitu/CCDM [43].

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

## Acknowledgements

We are grateful for the assistance of Ms Jasmine Lyall and Mr David Jessop for their time and effort in assisting with the human grader evaluations of real and generated retinal images. The study external validation data and compute was supported by INSIGHT, the Health Data Research Hub in Eye disease and Oculomics, which is affiliated with Health Data Research UK. The model development was funded by Diabetes UK via a Sir George Alberti research training fellowship grant to Dr Paul Nderitu (20/0006144). The external validation was funded by King's College Hospital Charity via a research grant to Dr Paul Nderitu and Professor Tim Jackson (D2312/102022/Jackson/991). The study was also supported by a Wellcome/EPSRC Centre for Medical Engineering grant (WT 203148/Z/16/Z) to Dr Christos Bergeles. Study funders did not have access to the study data, nor did they influence the study design, model development, data analysis, or manuscript preparation. The corresponding author had the final responsibility for the decision to submit the manuscript for publication.

## Author contributions

P.N., J.M.N., L.W., S.M., D.H., C.B., and T.J. formulated the study concept. P.N., J.M.N., C.B., and T.J. refined the study design. P.N., J.M.N., C.B., and T.J. created the protocol. L.W. performed the data extraction. P.N. accessed and verified the study data. P.N. performed the model development. P.N., J.M.N., S.M., C.B., and T.J. performed the primary data analysis. P.N. drafted the manuscript, with revisions and final approval performed by all authors.

## Competing interests

P.N., J.M.N., L.W., D.H., and C.B. have no conflicts of interest to declare. S.S.M. has received speaker and advisory board fees from Bayer and Allergan and research grants from Novartis. S.S.M. has assisted in a study validating the EyeArt v2.1 DLS for diabetic retinopathy detection. T.J. is an advisor to 2CTech, Alcon, Bayer, Dutch Ophthalmic Research Centre, iLumen, Opthea, Outlook Therapeutics, Oxurion, Pfizer, and Regeneron, has received conference support from Roche, and free equipment use for non-commercial trials from Zeiss, Oraya, and LKC. T.L.J., P.N., and D.H. employer, King's College Hospital, has received funding for participants enroled to multiple commercial clinical trials including for DR.
