## [Transparent Peer Review file · Communications Medicine]

Augmenting a prognostic deep learning system for referable diabetic retinopathy and maculopathy with synthetic retinal images

Corresponding Author: Dr Paul Nderitu

Version 0:

Reviewer comments:

Reviewer #1

(Remarks to the Author)

This manuscript explores the use of synthetic retinal images, to address the issues of labelled data scarcity and class imbalance, in improving diagnostic performance for referable diabetic retinopathy/maculopathy (DR/DME). In particular, with about 72,000 macula images, deep learning systems (DLS) were trained with various quantities of augmented images. It was found that internal test AUROC - but not external test AUROC - could be significantly improved with additional augmented images.

Some issues might be considered:

1. In the Study population and datasets subsection, it is stated that the macula field image was used. Is this the accepted practice for diabetic retinopathy/maculopathy classification, or was only the macula field image available? This might be briefly discussed.
2. In the Study population and datasets subsection, it is stated that age and diabetes duration were categorized into several groups. It might be briefly clarified as to whether the actual age/duration (to the year) was available and the categorization was performed post-hoc, or whether the values were recorded in the category groups. If the former, it might be discussed as to why the grouping was performed.
3. In the Model Development subsection, it is stated that the CCDM was used to generate 64x64 pixel synthetic retinal images. These 64x64 images were then upscaled to the original 512x512 size using a superresolution model conditioned on text and a prior lower-scale generated image. It might be clarified as to how relatively small-scale lesions and features of DR (such as microaneurysms) were represented - were they generated during the upscaling/superresolution process, and if so, how?
4. In the Prognostic DLS model development subsection, "×1 resampled real positive cases" might be briefly clarified - does this mean that the ratio of positive cases in training pDLS-R was twice that of the actual distribution (used for pDLS-N)? This appears to be the case from Table 2, but it could be considered to define it explicitly here for clarity.
5. In the CCDM synthetic retinal image quality and qualitative analysis subsection, certain characteristics of the generated synthetic images were discussed. It would be helpful if a small subset of the synthetic images were systematically analyzed by qualified graders/clinicians (e.g. as to whether synthetic images could be distinguished from real images, and whether the severity levels of DR/CME in the synthetic images reflected the values they were conditioned to generate, etc.) in a blind test.
6. In the CCDM development hyperparameters subsection of the Appendix, "classifier-free guidance" might be briefly elaborated upon. The hyperparameter optimization methodology (e.g. grid search) for the CCDM parameters such as guidance strength might also be stated.
7. The technical details of the superresolution model do not appear to be provided in either the main text or the Appendix.

Such details might be included, in particular on how conditioning was performed.

Reviewer #2

(Remarks to the Author)

This manuscript develops a conditional cascaded diffusion model (CCDM) to generate various types of synthetic retinal images, with the aim of reducing class imbalance. The study incorporates a large dataset with relatively complete information. Overall, the work in the paper is solid, but there are several issues that need revision:

- 1) The keywords are not well-chosen, as they are too scattered and disorganized.
- 2) How was the ratio of real images to synthetic images determined? Please provide a description.
- 3) The left and right eyes of the same patient were assigned to the training and validation datasets respectively. Could this cause bias in the results? Please provide a description.
- 4) The structure of the results section can be further optimized. Currently, it appears as just two large paragraphs, resulting in poor logic.
- 5) Figure 3 simply seems to demonstrate "the effect of clinicodemographic conditioning on synthetic retinal image generation". Could some descriptions and explanations be added?
- 6) As mentioned in the authors' discussion, most studies use GAN to generate synthetic images. What is the reason why CCDM is not widely used? Does the paper involve relevant technological innovations?

Reviewer #3

(Remarks to the Author)

1. Overall Structure & Presentation

The manuscript is well-structured with clear logical flow. Data presentation adheres to standard conventions, and key findings are effectively communicated.

2. Methodological Rigor & Significance

The experimental design is generally sound, and the results provide meaningful reference value for the target research domain.

3. Minor Language Revisions

Occasional grammatical inaccuracies require attention. For example:

- Line 291: "limited" → "limited to" (to ensure syntactic completeness).

4. Critical Suggestions for Enhancement

The feasibility and practical efficacy of using synthetic data generated by diffusion models to augment training of clinical decision-support models is a topic of intense research interests. To help reader better understand this study's impact, we recommend addressing the following:

- Distribution of limited positive cases in the development set of CCDM: Please clarify the clinicodemographic characteristics (e.g., age, sex, ethnicity...) of the positive cases used in CCDM training data.
- Computational cost of data generation: Quantify resources required for synthetic data production, including:
 - Hardware specifications (e.g., GPU type/memory)
 - Runtime duration (per dataset or per case)
- Cost implications for fine-grained control: As noted in Line 285, elaborate on the additional computational burden when implementing detailed image-level control in CCDM training/inference (e.g., 20% longer runtime? 2x GPU memory demand?).

Version 1:

Reviewer comments:

Reviewer #1

(Remarks to the Author)

We thank the authors for addressing our previous comments.

Reviewer #2

(Remarks to the Author)

Thanks for the authors' detailed reply. I have no more questions.

Reply to Reviewer Comments

Reviewer #1

Reviewer Summary

This manuscript explores the use of synthetic retinal images, to address the issues of labelled data scarcity and class imbalance, in improving diagnostic performance for referable diabetic retinopathy/maculopathy (DR/DME). In particular, with about 72,000 macula images, deep learning systems (DLS) were trained with various quantities of augmented images. It was found that internal test AUROC - but not external test AUROC - could be significantly improved with additional synthetic images.

Comments and Replies

Comment	Reply
1. In the Study population and datasets subsection, it is stated that the macula field image was used. Is this the accepted practice for diabetic retinopathy/maculopathy classification, or was only the macula field image available? This might be briefly discussed.	In the UK, both macula and optic disc-centred images are routinely used for DR screening. The macula field was selected for this study to isolate and evaluate the effect of synthetic augmentation on a single, clinically relevant field. This choice was also driven by computational constraints for CCDM training and sampling. However, two-field augmentation (as used in UK screening) is a natural next step for future work. Our focus for this study was on validating the efficacy of macula-specific augmentation in isolation. We have added the following to the manuscript discussion (line 358): “Computational constraints restricted experiments to single-field synthetic macula images as opposed to 2-field retinal images captured in UK DR screening. However, this constraint enabled the isolated evaluation of synthetic macula image augmentation.”
2. In the Study population and datasets subsection, it is stated that age and diabetes duration were categorized into several groups. It might be briefly clarified as to whether the actual age/duration (to the year) was available and the categorization was performed post-hoc, or whether the values were recorded in the category groups. If the former, it might be discussed as to why the grouping was performed.	Age and diabetes duration were grouped from continuous values to create a more consistent, clinically interpretable conditioning signal for the CCDM and we hypothesised this would also give a more detectable change to synthetic image features. We have amended the study population subsection (line 136) as follows: “Age and diabetes duration were categorised from continuous values to provide a more consistent, discrete conditioning signal for the CCDM, but using clinically relevant intervals.”

3. In the Model Development subsection, it is stated that the CCDM was used to generate 64x64 pixel synthetic retinal images. These 64x64 images were then upscaled to the original 512x512 size using a superresolution model conditioned on text and a prior lower-scale generated image. It might be clarified as to how relatively small-scale lesions and features of DR (such as microaneurysms) were represented - were they generated during the upscaling/superresolution process, and if so, how?	Microaneurysms and other subtle DR features which may not feature in the initial 64x64 image can be introduced during the super resolution step via the CCDM's conditioning on the presence/absence of mild DR as this would imply microaneurysms are present. Both the low-resolution image and text conditioning would influence the super resolution step. We do agree that small features could be more challenging to generate given the upscaling approach, but it is also possible the model has found some more subtle nonobvious encoding of these features which may not be readily apparent in the low-resolution image which get transmitted at up resolution. The high similarity of real and synthetic image embeddings (at 512 by 512) in UMAP suggesting that features are well-preserved through the generative pipeline. We also used a simple categorical to vector embedding as conditioning but more complex conditioning using CLIP or other techniques could also help CCDM generate these features more readily. In the manuscript we discuss this as follows (line 360): “Conditioning was achieved using a simple embedding of categorical clinicodemographic variables, but more complex caption-based conditioning using pre-trained CLIP embeddings or large language encoder models, could provide more nuanced control during synthetic retinal image generation of more subtle features such as exudates and microaneurysms.”
4. In the Prognostic DLS model development subsection, "x1 resampled real positive cases" might be briefly clarified - does this mean that the ratio of positive cases in training pDLS-R was twice that of the actual distribution (used for pDLS-N)? This appears to be the case from Table 2, but it could be considered to define it explicitly here for clarity.	Yes, the additional x1 resampled positive cases does mean twice the number of positive cases since each positive case is resampled again once. We have amended the model development section as follows (line 166): “(2) an augmented development dataset with double the number of positive cases of the unaugmented dataset achieved via x1 resampling of real positive cases (pDLS-R)”
5. In the CCDM synthetic retinal image quality and qualitative analysis subsection, certain characteristics of the generated synthetic images were discussed. It would be helpful if a small subset of the synthetic images were systematically analyzed by qualified graders/clinicians (e.g. as to whether synthetic images could be distinguished from real images, and whether the severity levels of DR/CME in the synthetic images reflected the values they were conditioned to generate, etc.) in a blind test.	We thank the reviewer for this very helpful suggestion, and we have conducted a Turing type test to evaluate the images qualitatively with a senior and junior human grader. We have removed prior points on Turing tests / human grader evaluations in the limitations section, now we have added this experiment. We had added the following to the methods/outcomes (line 184): “We conducted a human grader test where 50 real and 50 synthetic macula images were randomised and assessed by a senior and junior human grader from SEL-DESP. Their tasks were to grade: 1) The level of DR, and 2) classify images as real or generated. Graders were blinded to the proportion of real versus generated images and the level of DR (R0, R1, R2 or R3A) or maculopathy (M0 or M1). However, all images in the test had either no DR (R0) or mild

	DR (R1), and none had maculopathy (M0). Graders were also asked to provide the subjective criteria they devised to differentiate between real and generated images at the end of the test.” We have added the following to the results (line 230): “The senior grader achieved 100% accuracy in differentiating between real and generated retinal images. In contrast, the junior grader attained an overall accuracy rate of 62%, which included a 96% accuracy for real images and a 28% accuracy for generated images. The graders' accuracy in distinguishing between no DR and mild DR was 87.5%, comprising 95% accuracy for real images and 82% for generated images. Both graders accurately recognised the absence of maculopathy in all real and generated cases. Graders felt that synthetic images had unnaturally straight or angular blood vessels, fine vessels in periphery, and illogical vessel branching.” We have also added a paragraph to the discussion (line 279): “Although quantitative and qualitative metrics were favourable, evaluation by human graders revealed that the senior graders was generally able to distinguish synthetic images from real images, whereas a junior grader more frequently misclassified generated images as real. This contrasts with findings by a previous study by Kim et al., which reported no difference in real vs generated grading accuracy based on human grader experience.³⁰ A key distinction between their study and ours is that they used GAN-generated images and a training set composed primarily of normal eyes.³⁰ However, in concordance with our study, Kim et al., also reported that graders reported inconsistencies in retinal vasculature in generated retinal images.³⁰ Interestingly, one grader remarked that certain generated images elicited a queasy, uneasy feeling, akin to the ‘uncanny valley’ effect noted with virtual faces.³¹ Real images were more representative of the true DR level compared to generated images which were consistently graded as showing no retinopathy when conditioned to generate mild DR. This may be due to challenges in the CCDM synthesising small DR lesions like microaneurysms.”
6. In the CCDM development hyperparameters subsection of the Appendix, "classifier-free guidance" might be briefly elaborated upon. The hyperparameter optimization methodology (e.g. grid search) for the CCDM parameters such as guidance strength might also be stated.	We thank the reviewer for this suggestion. We have elaborated on the hyperparameters of the CCDM in the supplementary as follows: “Classifier-Free Guidance is a technique that enables image generation to be controlled by conditioning inputs (e.g. prompts) without relying on classifier networks. It adjusts the noise-prediction process through a guidance scale, providing versatile control over the degree of adherence to the conditioning inputs.^{5,6}” “The chosen hyperparameters, including for guidance strength, evaluation steps, and sigma churn were informed by the hyperparameter sweeps and optimal values reported in the original Imagen and Karras reports.^{5,7} Due to computational limitations, conducting our own sweeps was not feasible.”
7. The technical details of the superresolution model do not appear to be provided in either the main text or the Appendix. Such details might be included, in	Thank you for the feedback. To clarify, the CCDM refers to the sequence of the base low-resolution model then the super resolution model. Therefore, the hyperparameters in the supplementary, unless otherwise respectively specified, applies to both the base low-resolution

particular on how conditioning was performed.	and super resolution model under the CCDM umbrella term. We have provided and updated Figure 2 which shows the stage of the super resolution model within the CCDM.
---	---

Reviewer #2

Reviewer Summary

This manuscript develops a conditional cascaded diffusion model (CCDM) to generate various types of synthetic retinal images, with the aim of reducing class imbalance. The study incorporates a large dataset with relatively complete information. Overall, the work in the paper is solid, but there are several issues that need revision.

Comments and Replies

Comment	Reply
1)The keywords are not well-chosen, as they are too scattered and disorganized.	Thank you for the feedback. We have amended the key words as follows: Dataset Augmentation, Deep Learning, Artificial Intelligence, Diabetic Retinopathy, Diabetic Maculopathy, Synthetic Images, Prediction, Prognostic Modelling, Conditional Diffusion Models, Real images, Retinal Imaging
2)How was the ratio of real images to synthetic images determined? Please provide a description.	For our use case of class imbalance, we selected progressively higher proportions of real to generated images to reduce the ratio of positive to negative from 1:60 in the original dataset to approximately 1:10. The objective was not to achieve full class parity, but rather to mitigate the extent of imbalance for a modest computational sampling budget. We have clarified this in methods (Line 169) as follows: "We aimed to reduce the degree of class imbalance by a factor of ~5 which would incur a modest CCDM sampling computational cost compared to generating enough positive cases to match every negative case to get a full a fully balanced dataset" .
3)The left and right eyes of the same patient were assigned to the training and validation datasets respectively. Could this cause bias in the results? Please provide a description.	To clarify, left and right eyes were assigned either to training OR validation but not both, which, as you suggest, would bias the results. We have clarified this in the moths/datasets section as follows (Line 142): "The right and left eyes of the same patient were used either in the training or testing sets but not both."
4)The structure of the results section can be further optimized. Currently, it appears as just two large paragraphs, resulting in poor logic.	Thank you for the suggestion. We have added subheadings in various sections of the results and methods to make the manuscript clearer.
5)Figure 3 simply seems to demonstrate "the effect of clinicodemographic conditioning on synthetic	Thank you for this helpful suggestion. We have added a descriptive caption for Figure 3.

retinal image generation". Could some descriptions and explanations be added?	
6)As mentioned in the authors' discussion, most studies use GAN to generate synthetic images. What is the reason why CCDM is not widely used? Does the paper involve relevant technological innovations?	We thank the reviewer for their comments and provide justification below. While GANs have traditionally been preferred in medical and retinal imaging due to their established architectures, newer research shows diffusion models now achieve higher image fidelity and are surpassing GANs, which can be unstable in training. Conditional diffusion models are relatively new in medical imaging, as diffusion models were initially used for general image generation. The field is quickly advancing, with recent efforts adapting these models for synthetic retinal image creation. Our study is among the first to use clinicodemographic and a time dependant conditioning variable (future disease occurrence) to generate synthetic retinal images and augment a prognostic development dataset. However, model architecture (CCDM), is derived from a generalist CCDM (IMAGEN by Google), and we use open-access code https://github.com/lucidrains/imagen-pytorch.

Reviewer #3

Reviewer Summary

The manuscript is well-structured with clear logical flow. Data presentation adheres to standard conventions, and key findings are effectively communicated. The experimental design is generally sound, and the results provide meaningful reference value for the target research domain. The feasibility and practical efficacy of using synthetic data generated by diffusion models to augment training of clinical decision-support models is a topic of intense research interests. To help reader better understand this study's impact, we recommend addressing the following:

Comments and Replies

Comment	Reply
Occasional grammatical inaccuracies require attention. For example: - Line 291: "*limited*" → "*limited to*" (to ensure syntactic completeness).	Thank you for the helpful feedback - this error has now been corrected.
- Distribution of limited positive cases in the development set of CCDM: Please clarify the	We agree this is a valuable addition and have added the distribution of the characteristics for the positive training set cases to Table 1.

clinicodemographic characteristics (e.g., age, sex, ethnicity...) of the positive cases used in CCDM training data.	
- Computational cost of data generation: Quantify resources required for synthetic data production, including:  • Hardware specifications (e.g., GPU type/memory) • Runtime duration (per dataset or per case) 	We thank the reviewer for this helpful suggestion. We provide more details on the GPU memory and runtimes for CCDM training and inference in the methods (line 156) as follows: “CCDM were trained on two P6000 24GB NVIDIA GPUs with a total training time of 9 days (~50 epochs). The time taken to generate 9,071 test set images with a batch size of 8 was 9 hours and 50 minutes on a single P6000 (~31 seconds/iteration).”
- Cost implications for fine-grained control: As noted in Line 285, elaborate on the additional computational burden when implementing detailed image-level control in CCDM training/inference (e.g., 20% longer runtime? 2x GPU memory demand?).	Thank you for the useful suggestion. We have added to the methods to further elaborate on the computation cost together with the prior suggestion (line 158): “The use of conditioning to allow for the fine-grained control of image synthesis increases training time by ~20% and doubled the inference time.”